# Dissecting Recall of Factual Associations in Auto-Regressive Language Models

**Mor Geva**[1]     **Jasmijn Bastings**[1]     **Katja Filippova**[1]     **Amir Globerson**[2,3]

[1]Google DeepMind     [2]Tel Aviv University     [3]Google Research

{pipek, bastings, katjaf, amirg}@google.com

## Abstract

Transformer-based language models (LMs) are known to capture factual knowledge in their parameters. While previous work looked into *where* factual associations are stored, only little is known about *how* they are retrieved internally during inference. We investigate this question through the lens of information flow. Given a subject-relation query, we study how the model aggregates information about the subject and relation to predict the correct attribute. With interventions on attention edges, we first identify two critical points where information propagates to the prediction: one from the relation positions followed by another from the subject positions. Next, by analyzing the information at these points, we unveil a three-step internal mechanism for attribute extraction. First, the representation at the last-subject position goes through an enrichment process, driven by the early MLP sublayers, to encode many subject-related attributes. Second, information from the relation propagates to the prediction. Third, the prediction representation "queries" the enriched subject to extract the attribute. Perhaps surprisingly, this extraction is typically done via attention heads, which often encode subject-attribute mappings in their parameters. Overall, our findings introduce a comprehensive view of how factual associations are stored and extracted internally in LMs, facilitating future research on knowledge localization and editing.[1]

## 1 Introduction

Transformer-based language models (LMs) capture vast amounts of factual knowledge (Roberts et al., 2020; Jiang et al., 2020), which they encode in their parameters and recall during inference (Petroni et al., 2019; Cohen et al., 2023). While recent works focused on identifying *where* factual

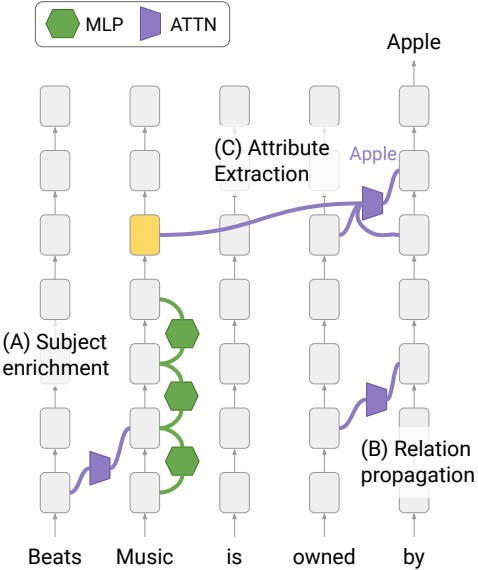

Figure 1: Illustration of our findings: given subject-relation query, a subject representation is constructed via attributes' enrichment from MLP sublayers (A), while the relation propagates to the prediction (B). The attribute is then extracted by the MHSA sublayers (C).

knowledge is encoded in the network (Meng et al., 2022a; Dai et al., 2022; Wallat et al., 2020), it remains unclear *how* this knowledge is extracted from the model parameters during inference.

In this work, we investigate this question through the lens of information flow, across layers and input positions (Elhage et al., 2021). We focus on a basic information extraction setting, where a subject and a relation are given in a sentence (e.g. *"Beats Music is owned by"*), and the next token is the corresponding attribute (i.e. *"Apple"*). We restrict our analysis to cases where the model predicts the correct attribute as the next token, and set out to understand how internal representations evolve across the layers to produce the output.

Focusing on modern auto-regressive decoder-only LMs, such an extraction process could be

---

[1]Our code is publicly available at `https://github.com/google-research/google-research/tree/master/dissecting_factual_predictions`

implemented in many different ways. Informally, the model needs to "merge" the subject and relation in order to be able to extract the right attribute, and this merger can be conducted at different layers and positions. Moreover, the attribute extraction itself could be performed by either or both of the multi-head self-attention (MHSA) and MLP sublayers.

To investigate this, we take a reverse-engineering approach, inspired by common genetic analysis methods (Griffiths et al., 2005; Tymms and Kola, 2008) and the recent work by Wang et al. (2022). Namely, we artificially block, or "knock out", specific parts in the computation to observe their importance during inference. To implement this approach in LLMs, we intervene on the MHSA sublayers by blocking the last position from attending to other positions at specific layers. We identify two consecutive critical points in the computation, where representations of the relation and then the subject are incorporated into the last position: first the relation and then the subject.

Next, to identify where attribute extraction occurs, we analyze the information that propagates at these critical points and the representation construction process that precedes them. This is done through additional interventions to the MHSA and MLP sublayers and projections to the vocabulary (Dar et al., 2022; Geva et al., 2022b; Nostalgebraist, 2020). We discover an internal mechanism for attribute extraction that relies on two key components. First, a *subject enrichment process*, through which the model constructs a representation at the last subject-position that encodes many subject-related attributes. Moreover, we find that out of the three sources that build a representation (i.e., the MHSA and MLP sublayers and the input token embeddings (Mickus et al., 2022)), the early MLP sublayers are the primary source for subject enrichment.

The second component is an *attribute extraction operation* carried out by the upper MHSA sublayers. For a successful extraction, these sublayers rely on information from both the subject representation and the last position. Moreover, extraction is performed by attention heads, and our analysis shows that these heads often encode subject-attribute mappings in their parameters. We observed this extraction behavior in ∼70% of the predictions.

Our analysis provides a significantly improved understanding of the way factual predictions are formed. The mechanism we uncover can be intuitively described as the following three key steps

(Fig. 1). First, information about the subject is enriched in the last subject token, across early layers of the model. Second, the relation is passed to the last token. Third, the last token uses the relation to extract the corresponding attribute from the subject representation, and this is done via attention head parameters. Unlike prior works on factual knowledge representation, which focus on mid-layer MLPs as the locus of information (e.g. Meng et al. (2022b)), our work highlights the key role of lower MLP sublayers and of the MHSA parameters. More generally, we make a substantial step towards increasing model transparency, introducing new research directions for knowledge localization and model editing.

## 2 Background and Notation

We start by providing a detailed description of the transformer inference pass, focusing on auto-regressive decoder-only LMs. For brevity, bias terms and layer normalization (Ba et al., 2016) are omitted, as they are nonessential for our analysis.

A transformer-based LM first converts an input text to a sequence $t_1, ...t_N$ of $N$ tokens. Each token $t_i$ is then embedded as a vector $\mathbf{x}_i^0 \in \mathbb{R}^d$ using an embedding matrix $E \in \mathbb{R}^{|\mathcal{V}| \times d}$, over a vocabulary $\mathcal{V}$. The input embeddings are then transformed through a sequence of $L$ transformer layers, each composed of a multi-head self-attention (MHSA) sublayer followed by an MLP sublayer (Vaswani et al., 2017). Formally, the representation $\mathbf{x}_i^\ell$ of token $i$ at layer $\ell$ is obtained by:

$$\mathbf{x}_i^\ell = \mathbf{x}_i^{\ell-1} + \mathbf{a}_i^\ell + \mathbf{m}_i^\ell \qquad (1)$$

where $\mathbf{a}_i^\ell$ and $\mathbf{m}_i^\ell$ are the outputs from the $\ell$-th MHSA and MLP sublayers (see below), respectively. An output probability distribution is obtained from the final layer representations via a prediction head $\delta$:

$$\mathbf{p}_i = \text{softmax}\big(\delta(\mathbf{x}_i^L)\big), \qquad (2)$$

that projects the representation to the vocabulary space, either through multiplying by the embedding matrix (i.e., $\delta(\mathbf{x}_i^L) = E\mathbf{x}_i^L$) or by using a trained linear layer (i.e., $\delta(\mathbf{x}_i^L) = W\mathbf{x}_i^L + \mathbf{u}$ for $W \in \mathbb{R}^{|\mathcal{V}| \times d}, \mathbf{u} \in \mathbb{R}^{|\mathcal{V}|}$).

**MHSA Sublayers** The MHSA sublayers compute *global* updates that aggregate information from all the representations at the previous layer. The $\ell$-th MHSA sublayer is defined using four

parameter matrices: three projection matrices $W_Q^\ell, W_K^\ell, W_V^\ell \in \mathbb{R}^{d \times d}$ and an output matrix $W_O^\ell \in \mathbb{R}^{d \times d}$. Following Elhage et al. (2021); Dar et al. (2022), the columns of each projection matrix and the rows of the output matrix can be split into $H$ equal parts, corresponding to the number of attention heads $W_Q^{\ell,j}, W_K^{\ell,j}, W_V^{\ell,j} \in \mathbb{R}^{d \times \frac{d}{H}}$ and $W_O^{\ell,j} \in \mathbb{R}^{\frac{d}{H} \times d}$ for $j \in [1, H]$. This allows describing the MHSA output as a sum of matrices, each induced by a single attention head:

$$\mathbf{a}_i^\ell = \sum_{j=1}^{H} A^{\ell,j} \left( X^{\ell-1} W_V^{\ell,j} \right) W_O^{\ell,j} \tag{3}$$

$$:= \sum_{j=1}^{H} A^{\ell,j} \left( X^{\ell-1} W_{VO}^{\ell,j} \right) \tag{4}$$

$$A^{\ell,j} = \gamma \left( \frac{\left( X^{\ell-1} W_Q^{\ell,j} \right) \left( X^{\ell-1} W_K^{\ell,j} \right)^T}{\sqrt{d/H}} + M^{\ell,j} \right) \tag{5}$$

where $X^\ell \in \mathbb{R}^{N \times d}$ is a matrix with all token representations at layer $\ell$, $\gamma$ is a row-wise softmax normalization, $A^{\ell,j} \in \mathbb{R}^{N \times N}$ encodes the weights computed by the $j$-th attention head at layer $\ell$, and $M^{\ell,j}$ is a mask for $A^{\ell,j}$. In auto-regressive LMs, $A^{\ell,j}$ is masked to a lower triangular matrix, as each position can only attend to preceding positions (i.e. $M_{rc}^{\ell,j} = -\infty \ \forall c > r$ and zero otherwise). Importantly, the cell $A_{rc}^{\ell,j}$ can viewed as a weighted edge from the $r$-th to the $c$-th hidden representations at layer $\ell - 1$.

**MLP Sublayers** Every MLP sublayer computes a *local* update for each representation:

$$\mathbf{m}_i^\ell = W_F^\ell \, \sigma \left( W_I^\ell \left( \mathbf{a}_i^\ell + \mathbf{x}_i^{\ell-1} \right) \right) \tag{6}$$

where $W_I^\ell \in \mathbb{R}^{d_i \times d}$ and $W_F^\ell \in \mathbb{R}^{d \times d_i}$ are parameter matrices with inner-dimension $d_i$, and $\sigma$ is a nonlinear activation function. Recent works has shown that transformer MLP sublayers can be cast as key-value memories (Geva et al., 2021) that store factual knowledge (Dai et al., 2022; Meng et al., 2022a).

## 3 Experimental Setup

We focus on the task of factual open-domain questions, where a model needs to predict an attribute $a$ of a given subject-relation pair $(s, r)$. A triplet $(s, r, a)$ is typically expressed in a question-answering format (e.g. *"What instrument did Elvis Presley play?"*) or as a fill-in-the-blank query (e.g. *"Elvis Presley played the ____"*). While LMs often succeed at predicting the correct attribute for such queries (Roberts et al., 2020; Petroni et al., 2019), it is unknown how attributes are extracted internally.

For a factual query $q$ that expresses the subject $s$ and relation $r$ of a triplet $(s, r, a)$, let $t = (t_1, ..., t_N)$ be the representation of $q$ as a sequence of tokens, based on some LM. We refer by the *subject tokens* to the sub-sequence of $t$ that corresponds to $s$, and by the *subject positions* to the positions of the subject tokens in $t$. The non-subject tokens in $q$ express the relation $r$.

**Data** We use queries from COUNTERFACT (Meng et al., 2022a). For a given model, we extract a random sample of queries for which the model predicts the correct attribute. In the rest of the paper, we refer to the token predicted by the model for a given query $q$ as the attribute $a$, even though it could be a sub-word and thus only the prefix of the attribute name (e.g. Wash for "Washington").

**Models** We analyze two auto-regressive decoder-only GPT LMs with different layouts: GPT-2 (Radford et al., 2019) ($L = 48$, 1.5B parameters) and GPT-J (Wang and Komatsuzaki, 2021) ($L = 28$, 6B parameters). Both models use a vocabulary with ~50K tokens. Also, GPT-J employs parallel MHSA and MLP sublayers, where the output of the $\ell$-th MLP sublayer for the $i$-th representation depends on $\mathbf{x}_i^{\ell-1}$ rather than on $\mathbf{a}_i^\ell + \mathbf{x}_i^{\ell-1}$ (see Eq. 6). We follow the procedure by Meng et al. (2022a) to create a data sample for each model, resulting in 1,209 queries for GPT-2 and 1,199 for GPT-J.

## 4 Overview: Experiments & Findings

We start by introducing our attention blocking method and apply it to identify critical information flow points in factual predictions (§5) – one from the relation, followed by another from the subject. Then, we analyze the evolution of the subject representation in the layers preceding this critical point (§6), and find that it goes through an enrichment process driven by the MLP sublayers, to encode many subject-related attributes. Last, we investigate how and where the right attribute is extracted from this representation (§7), and discover that this is typically done by the upper MHSA sublayers, via attention heads that often encode a subject-attribute mapping in their parameters.

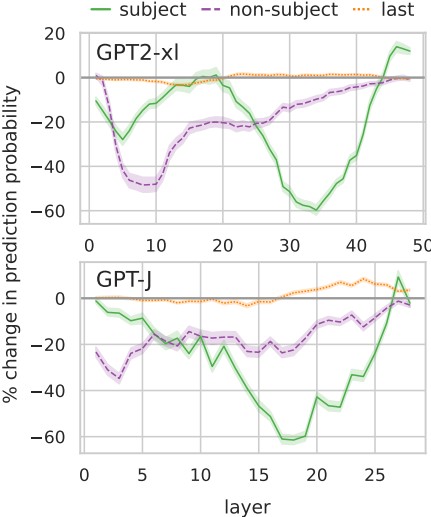

Figure 2: Relative change in the prediction probability when intervening on attention edges to the last position, for window sizes of 9 layers in GPT-2 and 5 in GPT-J.

## 5 Localizing Information Flow via Attention Knockout

For a successful attribute prediction, a model should process the input subject and relation such that the attribute can be read from the last position. We investigate how this process is done internally by "knocking out" parts of the computation and measuring the effect on the prediction. To this end, we propose a fine-grained intervention on the MHSA sublayers, as they are the only module that communicates information between positions, and thus any critical information must be transferred by them. We show that factual predictions are built in stages where critical information propagates to the prediction at specific layers during inference.

**Method: Attention Knockout**   Intuitively, critical attention edges are those that, when blocked, result in severe degradation in prediction quality. Therefore, we test whether critical information propagates between two hidden representations at a specific layer, by zeroing-out all the attention edges between them. Formally, let $r, c \in [1, N]$ such that $r \leq c$ be two positions. We block $\mathbf{x}_r^\ell$ from attending to $\mathbf{x}_c^\ell$ at a layer $\ell < L$ by updating the attention weights to that layer (Eq. 5):

$$M_{rc}^{\ell+1,j} = -\infty \quad \forall j \in [1, H] \qquad (7)$$

Effectively, this restricts the source position from obtaining information from the target position, at that particular layer. Notably, this is different from

causal tracing (Meng et al., 2022a), which checks what hidden representations restore the original prediction when given perturbed input tokens; we test where critical information *propagates* rather than where it is *located* during inference.

**Experiment**   We use Attention Knockout to test whether and, if so, where information from the subject and relation positions directly propagates to the last position. Let $\mathcal{S}, \mathcal{R} \subset [1, N)$ be the subject and non-subject positions for a given input. For each layer $\ell$, we block the attention edges from the last position to each of $\mathcal{S}$, $\mathcal{R}$ and the last ($N$-th) position, for a window of $k$ layers around the $\ell$-th layer, and measure the change in prediction probability. We set $k = 9$ (5) for GPT-2 (GPT-J).[2]

**Results**   Fig. 2 shows the results. For both GPT-2 and GPT-J, blocking attention to the subject tokens (solid green lines) in the middle-upper layers causes a dramatic decrease in the prediction probability of up to 60%. This suggests that critical information from the subject positions moves directly to the last position at these layers. Moreover, another substantial decrease of 35%-45% is observed for the non-subject positions (dashed purple lines). Importantly, critical information from non-subject positions precedes the propagation of critical information from the subject positions, a trend we observe for different subject-relation orders (§A.1). Example interventions are provided in §H.

Overall, this shows that there are specific disjointed stages in the computation with peaks of critical information propagating directly to the prediction from different positions. In the next section, we investigate the critical information that propagates from the subject positions to the prediction.

## 6 Intermediate Subject Representations

We saw that critical subject information is passed to the last position in the upper layers. We now analyze what information is contained in the subject representation at the point of transfer, and how does this information evolves across layers. To do this, we map hidden representations to vocabulary tokens via projection. Our results indicate that the subject representation contains a wealth of information about the subject at the point where it is transferred to the last position.

---

[2]Results for varying values of $k$ are provided in §A.4.

| | Subject | Example top-scoring tokens by the subject representation |
|---|---|---|
| | Iron Man | 3, 2, Marvel, Ultron, Avenger, comics, suit, armor, Tony, Mark, Stark, 2020 |
| GPT-2 | Sukarno | Indonesia, Buddhist, Thailand, government, Museum, Palace, Bangkok, Jakarta |
| | Roman Republic | Rome, Augustus, circa, conquered, fame, Antiqu, Greece, Athens, AD, Caesar |
| | Ferruccio Busoni | music, wrote, piano, composition, International, plays, manuscript, violin |
| GPT-J | Joe Montana | career, National, football, NFL, Award, retired, quarterback, throws, Field |
| | Chromecast | device, Audio, video, Wireless, HDMI, USB, Google, Android, technology, 2016 |

Table 1: Example tokens by subject representations of GPT-2 ($\ell = 40$) and GPT-J ($\ell = 22$).

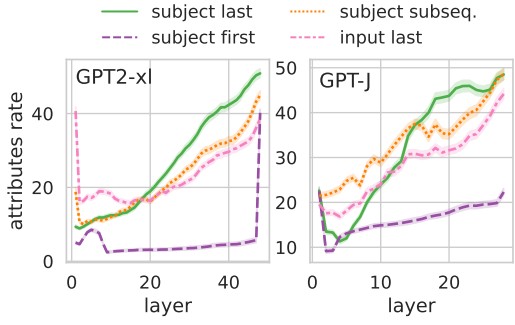

Figure 3: Attributes rate at different positions across layers (starting from layer 1), in GPT-2 and GPT-J.

### 6.1 Inspection of Subject Representations

**Motivating Observation**  To analyze what is encoded in a representation $\mathbf{h}_t^\ell$, we cast it as a distribution $\mathbf{p}_t^\ell$ over the vocabulary (Geva et al., 2022b; Nostalgebraist, 2020), using the same projection applied to final-layer representations (Eq. 2). Then, we inspect the $k$ tokens with the highest probabilities in $\mathbf{p}_t^\ell$. Examining these projections, we observed that they are informative and often encode several subject attributes (Tab. 1 and §D). Therefore, we turn to quantitatively evaluate the extent to which the representation of a subject encodes tokens that are semantically related to it.

**Evaluation Metric: Attributes Rate**  Semantic relatedness is hard to measure based on human judgment, as ratings are typically of low agreement, especially between words of various parts of speech (Zesch and Gurevych, 2010; Feng et al., 2017). Hence, we propose an automatic approximation of the subject-attribute relatedness, which is the rate of the predicted attributes in a given set of tokens known to be highly related to the subject. For a given subject $s$, we first create a set $\mathcal{A}_s$ of candidate attributes, by retrieving paragraphs about $s$ from Wikipedia using BM25 (Robertson et al., 1995), tokenizing each paragraph, and removing common words and sub-words. The set $\mathcal{A}_s$ consists

of non-common tokens that were mentioned in the context of $s$, and are thus likely to be its attributes. Further details on the construction of these sets are provided in §C. We define the *attributes rate* for a subject $s$ in a set of tokens $\mathcal{T}$ as the portion of tokens in $\mathcal{T}$ that appear in $\mathcal{A}_s$.

**Experiment**  We measure the attributes rate in the top $k = 50$ tokens by the *subject representation*, that is, the representation at the last-subject position, in each layer. We focus on this position as it is the only subject position that attends to all the subject positions, and thus it is likely to be the most critical (we validate this empirically in §A.3). We compare with the rate at other positions: the first subject position, the position after the subject, and the last input position (i.e., the prediction position).

**Results**  Fig. 3 shows the results for GPT-2 and GPT-J. In both models, the attributes rate at the last-subject position is increasing throughout the layers, and is substantially higher than at other positions in the intermediate-upper layers, reaching close to 50%. This suggests that, during inference, the model constructs attribute-rich subject representations at the last subject-position. In addition, critical information from these representations propagates to the prediction, as this range of layers corresponds to the peak of critical information observed by blocking the attention edges to the prediction (§5).

We have seen that the representation of the subject encodes many terms related to it. A natural question that arises is where these terms are extracted from to enrich that representation. In principle, there are three potential sources (Mickus et al., 2022), which we turn to analyze in the next sections: the static embeddings of the subject tokens (§6.2) and the parameters of the MHSA and MLP sublayers (§6.3).

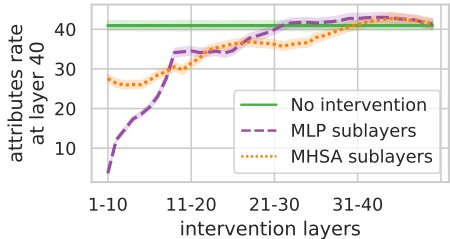

Figure 4: The attributes rate of the subject representation with and without canceling updates from the MLP and MHSA sublayers in GPT-2.

## 6.2 Attribute Rate in Token Embeddings

We test whether attributes are already encoded in the static embeddings of the subject tokens, by measuring the attributes rate, as in §6.1. Concretely, let $t_1, ..., t_{|s|}$ be the tokens representing a subject $s$ (e.g. Piet, ro, Men, nea for "Pietro Mennea"), and denote by $\bar{\mathbf{e}} := \frac{1}{|s|} \sum_{i=1}^{|s|} \mathbf{e}_{t_i}$ their mean embedding vector, where $\mathbf{e}_{t_i}$ is the embedding of $t_i$. We compute the attributes rate in the top $k = 50$ tokens by each of $\mathbf{e}_{t_i}$ and by $\bar{\mathbf{e}}$. We find that the highest attributes rate across the subject's token embeddings is 19.3 on average for GPT-2 and 28.6 in GPT-J, and the average rate by the mean subject embedding is 4.1 in GPT-2 and 11.5 in GPT-J. These rates are considerably lower than the rates by the subject representations at higher layers (Fig. 3). *This suggests that while static subject-token embeddings encode some factual associations, other model components are needed for extraction of subject-related attributes.*

## 6.3 Subject Representation Enrichment

We next assess how different sublayers contribute to the construction of subject representations through causal interventions.

**Method: Sublayer Knockout** To understand which part of the transformer layer "adds" the information about attributes to the representation, we simply zero-out the two key additive elements: the MHSA and MLP sublayers. Concretely, we zero-out updates to the last subject position from each MHSA and MLP sublayer, for 10 consecutive layers. Formally, when intervening on the MHSA (MLP) sublayer at layer $\ell$, we set $\mathbf{a}_i^{\ell'} = \mathbf{0}$ ($\mathbf{m}_i^{\ell'} = \mathbf{0}$) for $\ell' = \ell, ..., \min\{\ell + 9, L\}$ (see Eq. 1). For each intervention, we measure the effect on the attributes rate in the subject representation at some specific layer $\bar{\ell}$, where attribute rate is high.

**Results** Results with respect to layer $\bar{\ell} = 40$ in GPT-2 are shown in Fig. 4, showing that canceling the early MLP sublayers has a destructive effect on the subject representation's attributes rate, decreasing it by ~88% on average. In contrast, canceling the MHSA sublayers has a much smaller effect of <30% decrease in the attributes rate, suggesting that the MLP sublayers play a major role in creating subject representations. Results with respect to other layers and for GPT-J show similar trends and are provided in §C and §E, respectively. We further analyze this by inspecting the MLP updates, showing they promote subject-related concepts (§F).

Notably, these findings are consistent with the view of MLP sublayers as key-value memories (Geva et al., 2021; Dai et al., 2022) and extend recent observations (Meng et al., 2022a; Wallat et al., 2020) that factual associations are stored in intermediate layers, showing that they are spread across the early MLP sublayers as well.

## 7 Attribute Extraction via Attention

The previous section showed that the subject representation is enriched with information throughout the early-middle layers. But recall that in our prediction task, only one specific attribute is sought. How is this attribute extracted and at which point? We next show that (a) attribute extraction is typically carried out by the MHSA sublayers (§7.1) when the last position attends to the subject, (b) the extraction is non-trivial as it reduces the attribute's rank by the subject representation considerably (§7.2) and it depends on the subject enrichment process (§7.3), and (c) the relevant subject-attribute mappings are often stored in the MHSA parameters (§7.4). This is in contrast to commonly held belief that MLPs hold such information.

### 7.1 Attribute Extraction to the Last Position

Recall that the critical information from the subject propagates when its representation encodes many terms related to it, and after the critical information flow from the relation positions (§5). Thus, the last-position representation at this point can be viewed as a relation query to the subject representation. We therefore hypothesize that the critical information that flows at this point is the attribute itself.

**Experiment: Extraction Rate** To test this hypothesis, we inspect the MHSA updates to the last position in the vocabulary, and check whether the

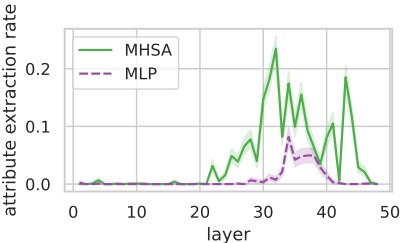

Figure 5: Attribute extraction rate across layers, for the MHSA and MLP sublayers in GPT-2.

|  | Extraction rate | # of extracting layers |
|---|---|---|
| MHSA | 68.2 | 2.15 |
| - all but subj. last + last | 44.4 | 1.03 |
| - all non-subj. but last | 42.1 | 1.04 |
| - last | 39.4 | 0.97 |
| - subj. last | 37.7 | 0.82 |
| - all but last | 32.9 | 0.55 |
| - subj. last + last | 32.8 | 0.7 |
| - non-subj. | 31.5 | 0.71 |
| - subj. | 30.2 | 0.51 |
| - all but subj. last | 23.5 | 0.44 |
| - all but first | 0 | 0.01 |
| MLP | 31.3 | 0.38 |

Table 2: Per-example extraction statistics across layers, for the MHSA and MLP sublayers, and MHSA with interventions on positions: (non-)subj. for (non-)subject positions, last (first) for the last (first) input position. Extraction rate here refers to the fraction of queries for which there was an extraction event in at least one layer.

top-token by each update matches the attribute predicted at the final layer. Formally, let

$$t^* := \arg\max(\mathbf{p}_N^L) \ ; \ t' := \arg\max(E\mathbf{a}_N^\ell)$$

be the token predicted by the model and the top-token by the $\ell$-th MHSA update to the last position (i.e., $\mathbf{a}_N^\ell$). We check the agreement between $t^*$ and $t'$ for every $\ell \in [1, L]$, and refer to agreement cases (i.e. when $t' = t^*$) as *extraction events*, since the attribute is being extracted by the MHSA. Similarly, we conduct the experiment while blocking the last position from attending to different positions (using attention knockout, see §5), and also apply it to the MLP updates to the last position.

**Results** Fig. 5 shows the extraction rate (namely the fraction of queries for which there was an extraction event) for the MHSA and MLP updates across layers in GPT-2, and Tab. 2 provides per-example extraction statistics (similar results for GPT-J are in §E). When attending to all the input positions, the upper MHSA sublayers promote the

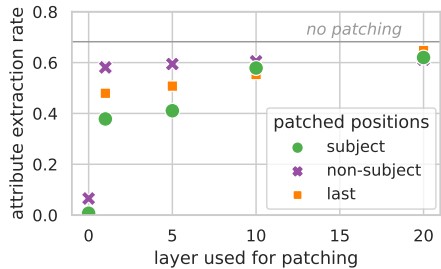

Figure 6: Extraction rate when patching representations from early layers at different positions in GPT-2.

attribute to the prediction (Fig. 5), with 68.2% of the examples exhibiting agreement events (Tab. 2). The layers at which extraction happens coincide with those where critical subject information propagates to the last position (Fig. 2), which further explains *why* this information is critical for the prediction.

Considering the knockout results in Tab. 2, attribute extraction is dramatically suppressed when blocking the attention to the subject positions (30.2%) or non-subject positions (31.5%). Moreover, this suppression is alleviated when allowing the last position to attend to itself and to the subject representation (44.4%), overall suggesting that critical information is centered at these positions.

Last, the extraction rate by the MLP sublayers is substantially lower (31.3%) than by the MHSA. Further analysis shows that for 17.4% of these examples, extraction by the MLP was preceded by an extraction by the MHSA, and for another 10.2% no extraction was made by the MHSA sublayers. *This suggests that both the MHSA and MLP implement attribute extraction, but MHSA is the prominent mechanism for factual queries.*

### 7.2 Extraction Significance

A possible scenario is that the attribute is already located at the top of the projection by the subject representation, and the MHSA merely propagates it "as-is" rather than extracting it. We show that this is not the case, by comparing the attribute's rank in the subject representation and in the MHSA update. For every extraction event with a subject representation $\mathbf{h}_s^\ell$, we check the attribute's rank in $\delta(\mathbf{h}_s^\ell)$, which indicates how prominent the extraction by the MHSA is (recall that, at an extraction event, the attribute's rank by the MHSA output is 1). We observe an average attribute rank of 999.5, which shows that the extraction operation promotes the specific attribute over many other candidate tokens.

## 7.3 Importance of Subject Enrichment

An important question is whether the subject representation enrichment is required for attribute extraction by the MHSA. Arguably, the attribute could have been encoded in early-layer representations or extracted from non-subject representations.

**Experiment** We test this by "patching" early-layer representations at the subject positions and measuring the effect on the extraction rate. For each layer $\ell = 0, 1, 5, 10, 20$, with 0 being the embeddings, we feed the representations at the subject positions as input to the MHSA at any succeeding layer $\ell' > \ell$. This simulates the MHSA operation at different stages of the enrichment process. Similarly, we patch the representations of the last position and of the other non-subject positions.

**Results** Results for GPT-2 are shown in Fig. 6 (and for GPT-J in §E). Patching early subject representations decreases the extraction rate by up to 50%, which stresses the importance of attributes enrichment for attribute recall. In contrast, patching of non-subject representations has a weaker effect, which implies that they are "ready" very early in the computation. These observations are further supported by a gradient-based feature attribution analysis (§B), which shows the influence of the early subject representations on the prediction.

Notably, for all the positions, a major increase in extraction rate is obtained in the first layer (e.g. $0.05 \rightarrow 0.59$ for non-subject positions), suggesting that the major overhead is done by the first layer.

## 7.4 "Knowledge" Attention Heads

We further investigate how the attribute is extracted by MHSA, by inspecting the attention heads' parameters in the embedding space and analyzing the mappings they encode for input subjects, using the interpretation by Dar et al. (2022).

**Analysis** To get the top mappings for a token $t$ by the $j$-th head at layer $\ell$, we inspect the matrix $W_{VO}^{\ell,j}$ in the embeddings space with

$$G^{\ell,j} := E^T W_{VO}^{\ell,j} E \in \mathbb{R}^{|V| \times |V|}, \qquad (8)$$

by taking the $k$ tokens with the highest values in the $t$-th row of $G^{\ell,j}$. Notably, this is an approximation of the head's operation, which is applied to contextualized subject representations rather than to token embeddings. For every extraction event with a subject $s$ and an attribute $a$, we then check if $a$ appears in the top-10 tokens for any of $s$'s tokens.

**Results** We find that for 30.2% (39.3%) of the extraction events in GPT-2 (GPT-J), there is a head that encodes the subject-attribute mapping in its parameters (see examples in §G). Moreover, these specific mappings are spread over 150 attention heads in GPT-2, mostly in the upper layers (24-45). Interestingly, further analysis of the frequent heads show they encode hundreds of such mappings, acting as "knowledge hubs" during inference (§G). *Overall, this suggests that factual associations are encoded in the MHSA parameters.*

## 8 Related Work

Recently, there has been a growing interest in knowledge tracing in LMs. A prominent thread focused on locating layers (Meng et al., 2022a; Wallat et al., 2020) and neurons (Dai et al., 2022) that store factual information, which often informs editing approaches (De Cao et al., 2021; Mitchell et al., 2022; Meng et al., 2022b). Notably, Hase et al. (2023) showed that it is possible to change an encoded fact by editing weights in other locations from where methods suggest this fact is stored, which highlights how little we understand about how factual predictions are built. Our work is motivated by this discrepancy and focuses on understanding the recall process of factual associations.

Our analysis also relates to studies of the prediction process in LMs (Voita et al., 2019; Tenney et al., 2019). Specifically, Haviv et al. (2023) used fine-grained interventions to show that early MLP sublayers are crucial for memorized predictions. Also, Hernandez et al. (2023) introduced a method for editing knowledge encoded in hidden representations. More broadly, our approach relates to studies of how LMs organize information internally (Reif et al., 2019; Hewitt and Manning, 2019).

Mechanistic interpretability (Olah, 2022; Nanda et al., 2023) is an emerging research area. Recent works used projections to the vocabulary (Dar et al., 2022; Geva et al., 2022b; Ram et al., 2022) and interventions in the transformer computation (Wang et al., 2022; Haviv et al., 2023) to study the inner-workings of LMs. A concurrent work by Mohebbi et al. (2023) studied contextualization in LMs by zeroing-out MHSA values, a method that effectively results in the same blocking effect as our *knockout* method. In our work, we leverage such methods to investigate factual predictions.

## 9 Conclusion

We carefully analyze the inner recall process of factual associations in auto-regressive transformer-based LMs, unveiling a core attribute extraction mechanism they implement internally. Our experiments show that factual associations are stored already in the lower layers in the network, and extracted eminently by the MLP sublayers during inference, to form attribute-rich subject representations. Upon a given subject-relation query, the correct attribute is extracted from these representations prominently through the MHSA sublayers, which often encode subject-attribute mappings in their parameters. These findings open new research directions for knowledge localization and model editing.

## Limitations

Some of our experiments rely on interpreting intermediate layer representations and parameters through projection to the vocabulary space. While this approach has been used widely in recent works (Geva et al., 2022b,a; Dar et al., 2022; Ram et al., 2022; Nostalgebraist, 2020), it only provides an approximation of the information encoded in these vectors, especially in early layers. In principle, this could have been an explanation to the increasing attributes rate in Fig. 3. However, this clear trend is unlikely to be explained only by this, given the low attribute rate at the embedding layer and the increase observed in the last few layers where approximation is better (Geva et al., 2021).

Another limitation is that our attention knockout intervention method does not account for "information leakage" across positions. Namely, if we block attention edges between two positions at a specific layer, it is still possible that information passed across these positions in earlier layers. For this reason, we block a range of layers rather than a single layer, which alleviates the possibility for such leakage. Moreover, our primary goal in this work was to identify critical attention edges, which are still critical even if such leakage occurs.

## Acknowledgements

We thank Asma Ghandeharioun for useful feedback and constructive suggestions.

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

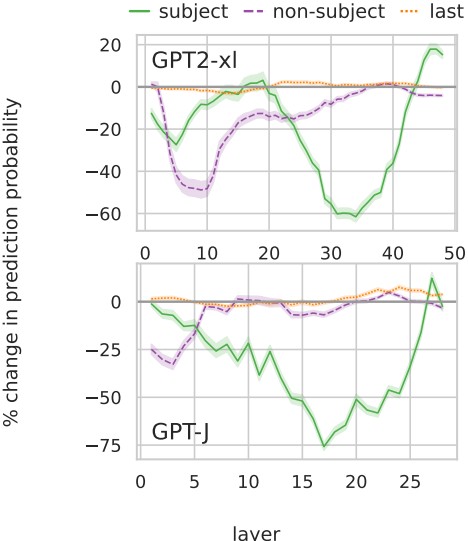

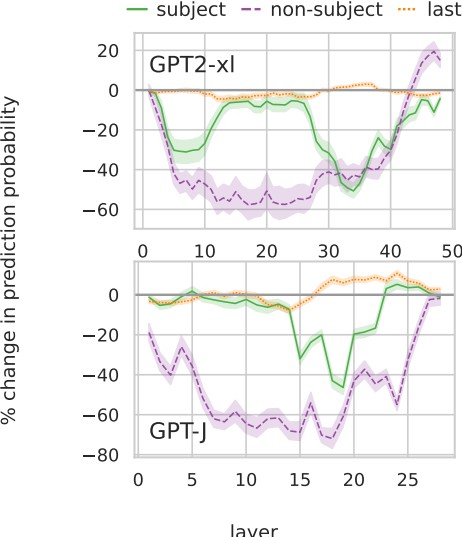

Figure 7: Relative change in the prediction probability when intervening on attention edges to the last position, for 9 layers in GPT-2 and 5 in GPT-J, for the subset of examples where the subject appears at the first position in the input.

Figure 8: Relative change in the prediction probability when intervening on attention edges to the last position, for 9 layers in GPT-2 and 5 in GPT-J, for the subset of examples where the subject appears after the first position in the input.

## A Additional Information Flow Analysis

### A.1 Subject-Relation Order

We break down the results in §5 by the subject-relation order in the input query, to evaluate whether we observe the same trends. Since the relation is expressed by all the non-subject tokens, we split the data into two subsets based on the the subject position: (a) examples where the subject appears in the first position (i.e. the subject appears before the relation), and (b) examples where it appears at later positions (i.e. the subject appears after the relation). We conduct the same attention blocking experiment as in §5, and show the results for the two subsets in Fig. 7 (a) and Fig. 8 (b).

In both figures, subject information passes to the last position at the same range of layers, showing that this observation hold in both cases. However, when the relation appears before the subject, blocking the attention to its positions has a more prominent impact on the prediction probability. Also, its effect is more spread-out across all the layers. We suggest that this different behavior is a result of a positional bias encoded in the first position in GPT-like models: regardless of which token appears in the first position, blocking attention to this position typically results in a substantial decrease in the prediction probability. The examples in §H demonstrate this. We further verify this in §A.2.

### A.2 First-position Bias

We observe that blocking the last position from attending to the first position, regardless of which token corresponds to it, has a substantial effect on the prediction probability (see examples in §H).

We quantify this observation and show that it does not change our main findings in §5. To this end, we conduct the same experiment in §5, but *without* blocking the attention edges to the first position. Results are provided in Fig. 9, showing the same trends as observed when blocking the attention to the first position as well; in both GPT-2 and GPT-J, there are clear peaks of critical information from subject and non-subject positions propagating to the last position at different layers, which when blocked reduce the prediction probability drastically.

Nonetheless, the decrease in probability at these peaks is smaller in magnitude when the first position is not blocked compared to when it is. For example, blocking the subject positions leads to a decrease of up to 40% when the last position can attend to the first position, compared to 60% when it cannot (Fig. 2). Likewise, blocking the non-subject positions (which correspond to the relation) leads to greater impact across the early-intermediate layers when the first position is blocked compared to when it is not. For instance, intervening on layers 5-20 in GPT-J constantly decreases the out-

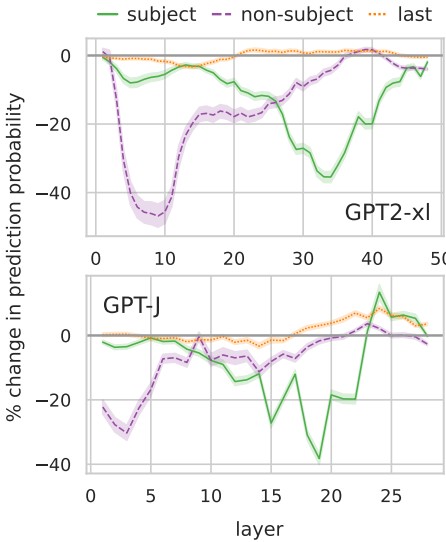

Figure 9: Relative change in the prediction probability when intervening on attention edges to the last position, for 9 layers in GPT-2 and 5 in GPT-J. Here, we do not block attention edges to the first position in the input.

put probability by ~20% when the first position is blocked (Fig. 2) compared to <5% when it is not.

### A.3 Information Flow from Subject Positions

We conjecture that, due to auto-regressivity, subject representations with critical information for the prediction are formed at the last-subject position. To verify that, we refine our interventions in §5, and block the attention edges to the last position from all the subject positions except one. We then measure the effect of these interventions on the prediction probability, which indicates from which position critical information propagates to the prediction.

Fig. 10 depicts the results for GPT-2, showing that indeed the prediction is typically damaged by $50\% - 100\%$ when the last subject-position is blocked (i.e. "first" and "before-last"), and usually remains intact when this position is not blocked (i.e. "last").

### A.4 Window Size

We examine the effect of the window size hyperparameter on our information flow analysis, as conducted in §5 for the last position in GPT-2 and GPT-J. Results of this analysis with varying window sizes of $k = 1, 5, 9, 13, 17, 21$ are provided in Fig. 11 for GPT-2 and Fig. 12 for GPT-J. Overall, the same observations are consistent across different window sizes, with two prominent sites

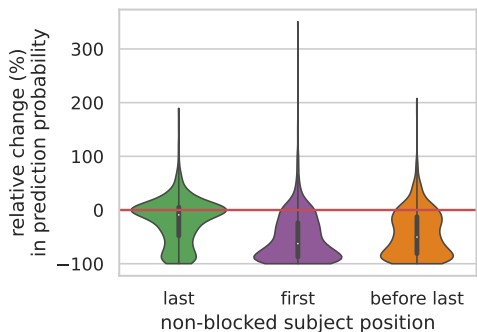

Figure 10: Relative change in the prediction probability when intervening on attention edges to the last position from different subject positions in GPT-2.

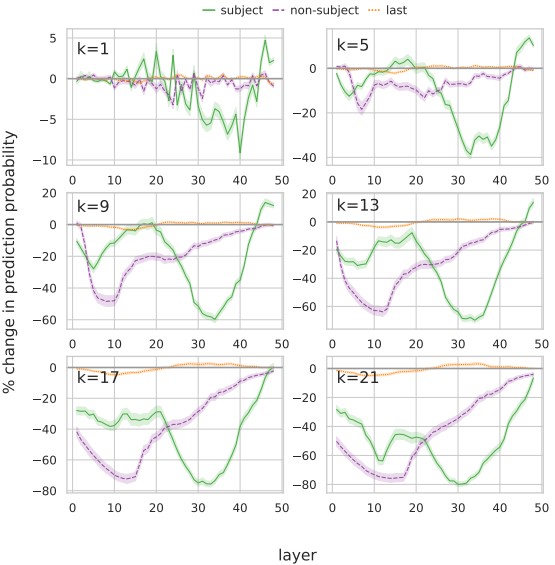

Figure 11: Relative change in the prediction probability when intervening on attention edges from different positions to the last position, for windows of $k$ layers in GPT-2. The plots show the analysis for varying values of $k$.

of critical information flow to the last position – one from the relation positions in the early layers, followed by another from the subject positions in the upper layers. An exception, however, can be observed when knocking out edges from the relation positions using just a single-layer window in GPT-2; in this case, no significant change in the prediction probability is apparent. This might imply that critical information from the relation positions is processed in multiple layers. Moreover, the decrease in the prediction probability becomes more prominent when for larger values of $k$. This is expected, as knocking out more attention edges in the computation prevents the model from contextualizing the input properly.

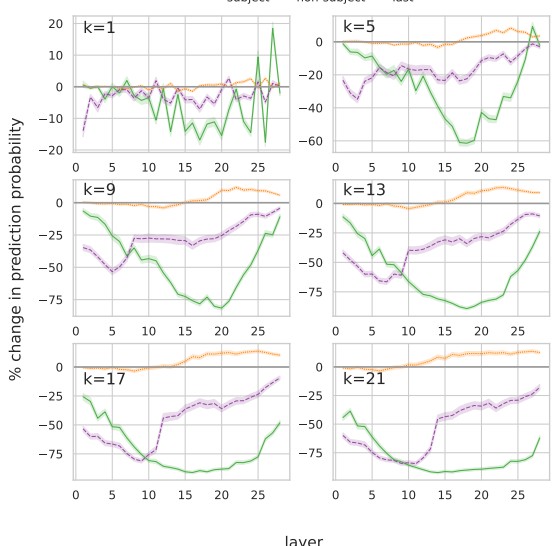

Figure 12: Relative change in the prediction probability when intervening on attention edges from different positions to the last position, for windows of $k$ layers in GPT-J. The plots show the analysis for varying values of $k$.

## B Gradient-based Analysis

Gradient-based feature attribution methods, also known as saliency methods, are a way to inspect what happens inside neural models for predictions of specific examples. Typically, they result in a heatmap over the input (sub)tokens, i.e., highlighted words. We distinguish between sensitivity and saliency (Ancona et al., 2019; Bastings and Filippova, 2020): methods such as Gradient-L2 (Li et al., 2016) show to which inputs the model is *sensitive*, i.e., where a small change in the input would make a large change in the output, but a method such as Gradient-times-Input (Denil et al., 2014) reflects *salience*: it shows (approximately) how much each input contributes to this particular logit value. The latter is computed as:

$$\nabla_{\mathbf{x}_i^\ell} f_c^\ell(\mathbf{x}_1^\ell, \mathbf{x}_2^\ell, \dots, \mathbf{x}_N^\ell) \odot \mathbf{x}_i^\ell \qquad (9)$$

where $\mathbf{x}_i^\ell$ is the input at position $i$ in layer $\ell$ (with $\ell = 0$ being the embeddings which are used for Gradient-times-Input), and $f_c^\ell(\cdot)$ the function that takes the input sequence at layer $\ell$ and returns the logit for the target/predicted token $c$. To obtain a scalar importance score for each token, we take the $L_2$ norm of each vector and normalize the scores to sum to one. For our analysis we use a generaliza-

tion of this method[3], and apply it not just to the input (sub)word embeddings ($\ell = 0$), but also to each intermediate transformer layer ($\ell = 1, \dots, N$).[4]

Fig. 13 shows the per-layer gradient-times-input analysis for the input "Beats Music is owned by" for GPT-2 and GPT-J. This analysis supports our earlier findings (§6, Fig. 6) that shows the "readiness" of subject vs. non-subject positions: We observe both for both models that the subject positions remain relevant until deep into the computation, while the subject is being enriched with associations. Moreover, the input for 'owned' is relevant for the prediction in the first few layers, after which the plot suggests it is incorporated into the final position. That final position becomes more relevant for the prediction the deeper we get into the network, as it seemingly incorporates information from other positions, before it is used as the point to make the final prediction for target 'Apple' at layer 48 (28). At that point, the final position has virtually all relevance, in line with what we would expect for an auto-regressive model.

In Fig. 14 we go beyond a single example, and show per-layer gradient-times-input aggregated over the dataset (a subset of COUNTERFACT, cf. §3) for each model. We can see that the observations we made for the single example in Fig. 13 also hold in general: the subject tokens remain relevant until about $\frac{2}{3}$ into the network depth, and the relation (indicated by "further tokens") is relevant in the first few layers, after which it becomes less relevant to the prediction.

## C Attributes Rate Evaluation

### C.1 Evaluation Details

We provide details on the construction process of candidate sets for attributes rate evaluation (§6.1). Given a subject $s$, first we use the BM25 algorithm (Robertson et al., 1995) to retrieve 100 paragraphs from the English Wikipedia[5] with $s$ being the query. From the resulting set, we keep only paragraphs for

---

[3]Concurrently, the use of per-layer gradient-times-input, also called gradient-times-activation, was also proposed by Sarti et al. (2023) for computing *contrastive* (Yin and Neubig, 2022) per-layer explanations.

[4]We add an `Identity` layer to the output of each transformer layer block, and capture the output and gradient there, after the attention and MLP sublayers have been added to the residual. This is important especially for GPT-J, where the attention and MLP sublayers are computed in parallel, so capturing the output and gradient only at the MLP would result in incorrect attributions.

[5]We use the dump of October 13, 2021.

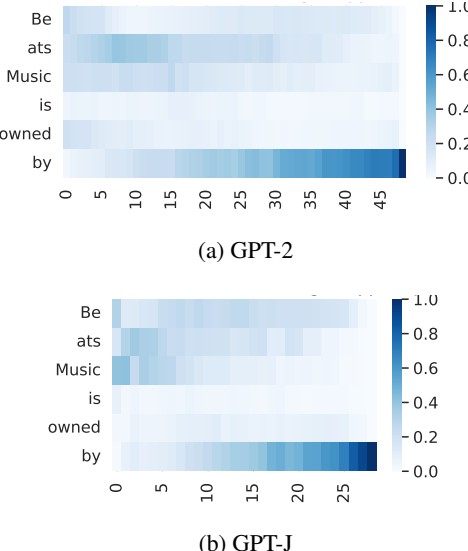

(a) GPT-2

(b) GPT-J

Figure 13: Gradient-times-Activation analysis for the example "Beats Music is owned by Apple".

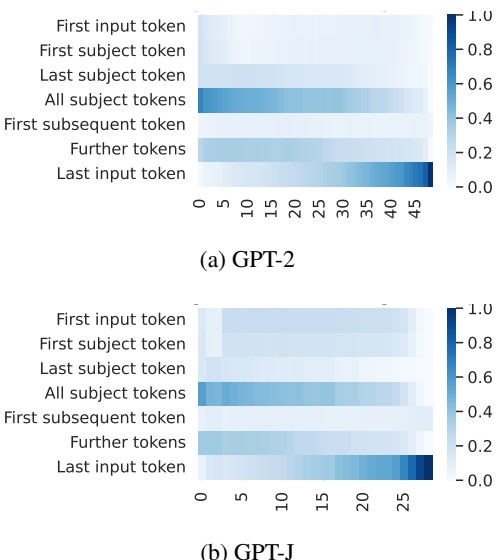

(a) GPT-2

(b) GPT-J

Figure 14: Gradient-times-Activation averaged over all examples.

which the subject appears as-is in their content or in the title of the page/section they are in. This is to avoid noisy paragraph that could be obtained from partial overlap with the subject (e.g. retrieving a paragraph about "2004 Summer Olympics" for the subject "2020 Summer Olympics"). This process results in 58.1 and 54.3 paragraphs per subject on average for the data subsets of GPT-2 and GPT-J, respectively.

Next, for each model, we tokenize the sets of paragraphs, remove duplicate tokens, and tokens with less than 3 characters (excluding spaces). The later is done to avoid tokens representing frequent short sub-words like S and 's. For stopwords removal, we use the list from the NLTK package.[6] This yields the final sets $\mathcal{A}_s$, of 1154.4 and 1073.1 candidate tokens on average for GPT-2 and GPT-J, respectively.

### C.2 Additional Sublayer Knockout Results

We extend our analysis in §6.3, where we analyzed the contribution of the MLP and MHSA sublayers to the subject enrichment process. Specifically, we now measure the effect of knocking out these sublayers on the attribute rate at *any* successive layer, rather than on a single upper layer. Results for GPT-2 are presented in Fig. 15 (MLP sublayers knockouts) and 16 (MHSA sublayers knockouts), showing similar trends where canceling updates from the MLP sublayers decreases the attributes rate dramatically, while a more benign effect is

---
[6]https://www.nltk.org/

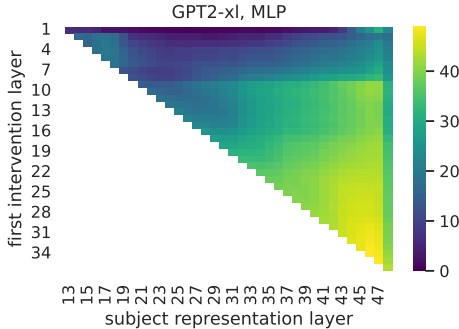

Figure 15: The attributes rate of the subject representation at different layers, when intervening on (canceling) different 10 consecutive MLP sublayers in GPT-2.

observed when canceling MHSA updates.

### D   Projection of Subject Representations

We provide additional examples for top-scoring tokens in the projection of subject representations across layers, in GPT-2 and GPT-J. Tab. 3 (Tab. 4) shows the tokens for the subject "Mark Messier" ("iPod Classic") across layers in GPT-2 and GPT-J, excluding stopwords and sub-word tokens.

### E   Additional Results for GPT-J

We supply here additional results for GPT-J. Fig. 17 shows the effect of canceling updates from the MHSA and MLP sublayers on the attributes rate of the subject representation at layer 22 (§6.3). Fig. 18 and Tab. 5 provide the extraction rate by the MHSA and MLP across layers and per-example extraction statistics, respectively (§7). Last, Fig. 19

| | $\ell$ | **Top-scoring tokens by the subject representation** |
|---|---|---|
| | 25 | `Jr, Sr, era, Era, MP, JR, senior, Clarence, final, stars, Junior, High, Architects` |
| GPT-2 | 35 | `Hockey, hockey, Jr, NHL, Rangers, Canucks, Sr, Islanders, Leafs, Montreal, Oilers` |
| | 40 | `NHL, Hockey, Jr, Canucks, retired, hockey, Toronto, Sr, Islanders, Leafs, jersey, Oilers` |
| | 13 | `Jr, Archives, ice, ring, International, Jersey, age, Institute, National, aged, career` |
| GPT-J | 17 | `hockey, NHL, Jr, Hockey, ice, Canadian, Canada, Archives, ice, Toronto, ring, National` |
| | 22 | `NHL, National, hockey, Jr, Foundation, Hockey, Archives, Award, ice, career, http` |

Table 3: Top-scoring tokens by the subject representations of "Mark Messier" across intermediate-upper layers ($\ell$) in GPT-2 and GPT-J.

| | $\ell$ | **Top-scoring tokens by the subject representation** |
|---|---|---|
| | 25 | `Edition, Ribbon, Series, Mini, version, Card, CR, XL, MX, XT, Cube, RX, Glow, speakers` |
| GPT-2 | 35 | `iPod, Bluetooth, Apple, Mini, iPhone, iOS, speaker, Edition, Android, Controller` |
| | 40 | `iPod, headphone, Bluetooth, speakers, speaker, Apple, iPhone, iOS, audio, headphones` |
| | 13 | `Series, device, Apple, iPod, model, Music, iPhone, devices, models, style, music` |
| GPT-J | 17 | `Series, model, device, style, Music, series, Archives, music, design, interface, models` |
| | 22 | `iPod, Music, music, song, Series, iPhone, Apple, songs, Music, review, series, Audio` |

Table 4: Top-scoring tokens by the subject representations of "iPod Classic" across intermediate-upper layers ($\ell$) in GPT-2 and GPT-J.

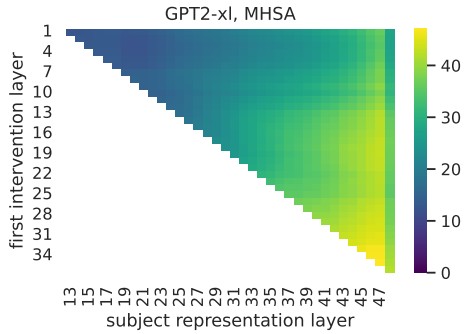

Figure 16: The attributes rate of the subject representation at different layers, when intervening on (canceling) different 10 consecutive MHSA sublayers in GPT-2.

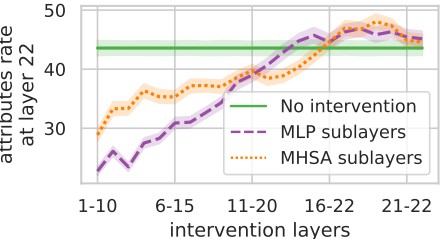

Figure 17: The attributes rate of the subject representation with and without canceling updates from the MLP and MHSA sublayers in GPT-J.

shows the effect of replacing intermediate representations with early layer representations at different positions on the attribute extraction rate (§7).

Overall, these results are consistent with those for GPT-2 described throughout the paper.

## F  Analysis of MLP Outputs

Following the observation that the early MLP sublayers are crucial for attribute enrichment of subject representations, we further analyze their updates to these representations. To this end, we decompose these updates into sub-updates that, according to Geva et al. (2022b), often encode human-interpretable concepts. Concretely, We decompose

Eq. 6 to a linear combination of parameter vectors of the second MLP matrix:

$$\mathbf{m}_i^{\ell} = \sum_{j=1}^{d_{inner}} \sigma\left(\mathbf{w}_I^{(\ell,j)}\left(\mathbf{a}_i^{\ell} + \mathbf{x}_i^{\ell-1}\right)\right) \cdot \mathbf{w}_F^{(\ell,j)},$$

where $d_{inner}$ is the MLP inner dimension, and $\mathbf{w}_I^{(\ell,j)}, \mathbf{w}_F^{(\ell,j)}$ are the $j$-th row and column of $W_I^{\ell}, W_F^{\ell}$, respectively. We then take the 100 vectors in $W_F^{\ell}$ with the highest contribution to this sum, for $\ell = 1, ..., 20$ in GPT-2, and inspect the top-scoring tokens in their projections to $E$.

From manual inspection, we indeed were able to identify cases where concepts related to the input subject are promoted by the dominant sub-updates. Examples are provided in Tab. 6. We note that quantifying this process is non-trivial (either with

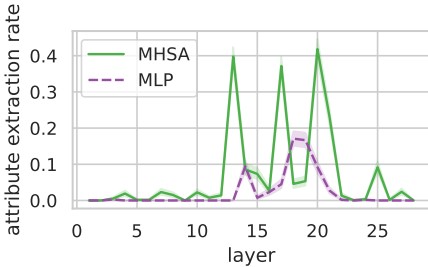

Figure 18: Attribute extraction rate across layers, for the MHSA and MLP sublayers in GPT-J.

|  | Extraction rate | # of extracting layers |
|---|---|---|
| MHSA | 76.7 | 1.9 |
| - last | 46.7 | 0.8 |
| - non-subj. | 45.6 | 0.8 |
| - all non-subj. but last | 44.4 | 0.9 |
| - subj. last | 41.1 | 0.7 |
| - subj. last + last | 40.3 | 0.7 |
| - all but subj. last + last | 34.8 | 0.5 |
| - all but subj. last | 31.3 | 0.5 |
| - subj. | 27.8 | 0.3 |
| - all but last | 24.8 | 0.3 |
| - all but first | 0 | 0 |
| MLP | 41 | 0.6 |

Table 5: Per-example extraction statistics across layers, for the MHSA and MLP sublayers, and MHSA with interventions on positions: (non-)subj. for (non-)subject positions, last (first) for the last (first) input position.

human annotations or automatic methods), and so we leave this for future work.

## G  Subject-Attribute Mappings in Attention Heads

In §7, we showed that for many extraction events, it is possible to identify specific heads that encode mappings between the input subject and predicted attribute in their parameters. We provide examples for such mappings in Tab. 7.

As a further analysis, we inspected the heads that repeatedly extract the attribute for different input queries in GPT-2. Specifically, for the matrix $W_{VO}^{\ell,j}$ of the $j$-th head at the $\ell$-th layer, we analyze its projection to the vocabulary $G^{\ell,j}$ (Eq. 8), but instead of looking at the top mappings for a specific subject-token (i.e. a row in $G^{\ell,j}$), we look at the top mappings across all rows, as done by (Dar et al., 2022). In our analysis, we focused on the heads that extract the attribute for at least 10% of the queries (7 heads in total). We observed that these heads act as "knowledge hubs", with their top mappings encoding hundreds of factual asso-

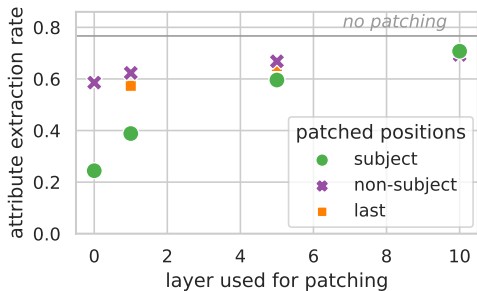

Figure 19: Extraction rate when patching representations from early layers at different positions in GPT-J.

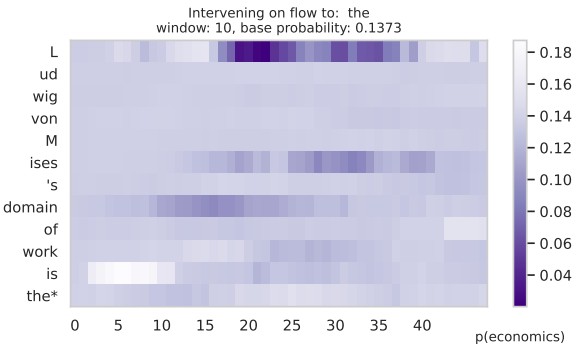

Figure 20: Example intervention on information flow to the last position of the input `Ludwig von Mises's domain of work is the`, using a window size 10, in GPT-2.

ciations that cover various relations. Tab. 8 shows example mappings by these heads.

## H  Example Interventions on Information Flow

Example interventions in GPT-2 are shown in Fig. 20, 21, 22, 23, 24, and in GPT-J in Fig. 25, 26, 27, 28, 29.

| Subject | Description | Layer, Dim. | Top tokens in the projection |
|---|---|---|---|
| United Launch Alliance | American spacecraft launch service provider | 1, 5394 | `jet, flights, aircraft, Cargo, passenger, plane, rockets, airspace, carrier, missiles, aerospace` |
| Yakuza 2 | Action-adventure video game | 2, 4817 | `Arcade, pixels, arcade, Wii, Fighters, pixels, Minecraft, Sega, Sonic, GPUs, Hardware, downloadable, multiplayer, Developers, livestream` |
| Leonard Bernstein | American composer, pianist, music educator, and author | 4, 248 | `violin, rehearsal, opera, pian, composer, Harmony, ensemble, Melody, piano, musicians, poets, Orchestra, Symphony` |
| Brett Hull | Canadian–American former ice hockey player | 4, 5536 | `discipl, athlet, Athletics, League, Hockey, ESPN, former, Sports, NHL, athleticism, hockey, Champions` |
| Bhaktisiddhanta Saraswati | Spiritual master, instructor, and revivalist in early 20th century India | 5, 3591 | `Pradesh, Punjab, Bihar, Gandhi, Laksh, Hindu, Hindi, Indian, guru, India, Tamil, Guru, Krishna, Bengal` |
| Mark Walters | English former professional footballer | 10, 1078 | `scorer, offence, scoring, defences, backfield, midfielder, midfield, striker, fielder, playoffs, rebounds, rushing, touchdowns` |
| Acura ILX | Compact car | 19, 179 | `Highway, bike, truck, automobile, Bicycle, motorists, cycle, Motor, freeway, vehicle, commute, Route, cars, motorcycle, Tire, streetcar, traffic` |
| Beats Music | Online music streaming service owned by Apple | 20, 5488 | `Technologies, Technology, Apple, iPod, iOS, cloud, Appl, engineering, software, platform, proprietary` |

Table 6: Top-scoring tokens in the projection of dominant MLP sub-updates to the subject representation in GPT-2.

| Matrix | Subject | Top mappings |
|---|---|---|
| $W_{OV}^{(33,13)}$ | *Barry Zito* | `Baseball, MLB, infield, Bethesda, RPGs, pitching, outfield, Iw,` **`pitcher`** |
| $W_{OV}^{(37,3)}$ | *iPod Classic* | `Macintosh, iPod, Mac, Perse,` **`Apple`**`, MacBook, Beck, Philipp` |
| $W_{OV}^{(44,16)}$ | *Philippus van Limborch* | `Dutch, Netherlands, van,` **`Amsterdam`**`, Holland, Vanessa` |

Table 7: Example mappings between subject-tokens (underlined) and attributes (**bold**) encoded in attention parameters (tokens for the same word are not shown).

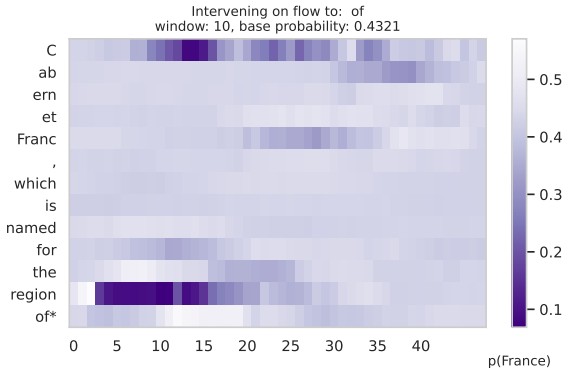

Figure 22: Example intervention on information flow to the last position of the input `Cabernet Franc, which is named for the region of`, using a window size 10, in GPT-2.

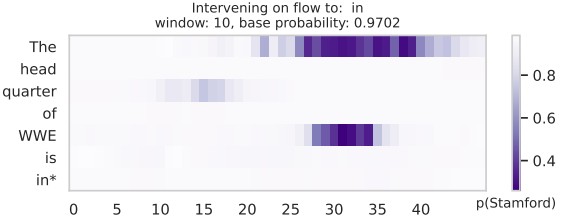

Figure 21: Example intervention on information flow to the last position of the input `The headquarter of WWE is in`, using a window size 10, in GPT-2.

| Matrix | Top mappings |
|---|---|
| $W_{OV}^{(43,25)}$ | (Finnish, Finland)
(Saskatchewan, Canadian)
(Finland, Finnish)
(Saskatchewan, Alberta)
(Helsinki, Finnish)
(Illinois, Chicago)
(Chicago, Illinois)
(Blackhawks, Chicago)
(Australia, Australians)
(NSW, Sydney)
(Minneapolis, Minnesota)
(Auckland, NZ)
(Texas, Texans) |
| $W_{OV}^{(36,20)}$ | (Japanese, Tokyo)
(Koreans, Seoul)
(Korean, Seoul)
(Haiti, Haitian)
(Korea, Seoul)
(Mexican, Hispanic)
(Spanish, Spanish)
(Japanese, Japan)
(Hawai, Honolulu)
(Dublin, Irish)
(Norwegian, Oslo)
(Israelis, Jewish)
(Vietnamese, Thai)
(Israel, Hebrew) |
| $W_{OV}^{(31,9)}$ | (oglu, Turkish)
(Tsuk, Japanese)
(Sven, Swedish)
(Tsuk, Tokyo)
(Yuan, Chinese)
(stadt, Germany)
(von, German)
(Hiro, Japanese)
(Sven, Norwegian)
(stadt, Berlin)
(Yuan, Beijing)
(Sven, Danish)
(oglu, Erdogan)
(Fei, Chinese)
(Samurai, Japanese) |

Table 8: Example top mappings encoded in the parameters of attention heads that extract the attribute for many input queries.

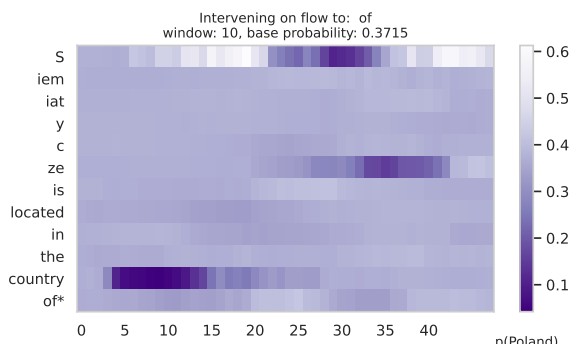

Figure 23: Example intervention on information flow to the last position of the input `Siemiatycze is located in the country of`, using a window size 10, in GPT-2.

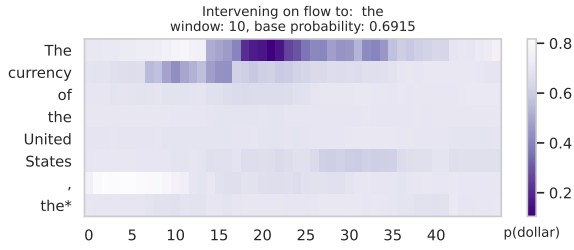

Figure 24: Example intervention on information flow to the last position of the input `The currency of the United States, the`, using a window size 10, in GPT-2.

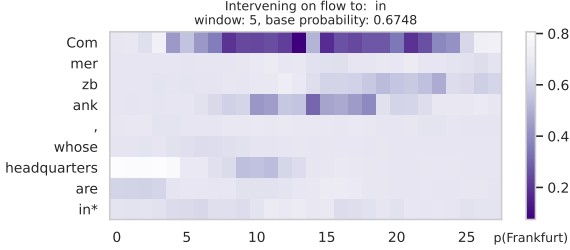

Figure 25: Example intervention on information flow to the last position of the input `Commerzbank, whose headquarters are in`, using a window size 5, in GPT-J.

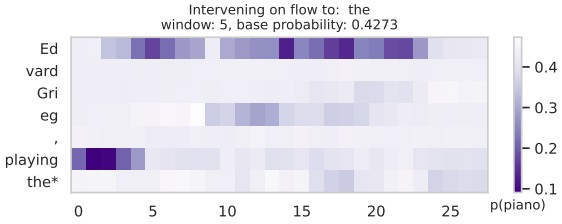

Figure 26: Example intervention on information flow to the last position of the input `Edvard Grieg, playing the`, using a window size 5, in GPT-J.

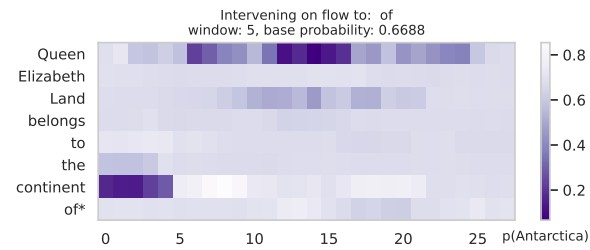

Figure 27: Example intervention on information flow to the last position of the input `Queen Elizabeth Land belongs to the continent of`, using a window size 5, in GPT-J.

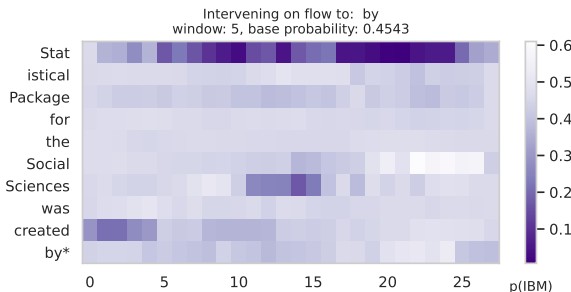

Figure 28: Example intervention on information flow to the last position of the input `Statistical Package for the Social Sciences was created by`, using a window size 5, in GPT-J.

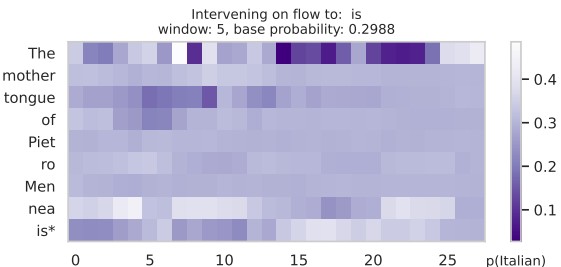

Figure 29: Example intervention on information flow to the last position of the input `The mother tongue of Pietro Mennea is`, using a window size 5, in GPT-J.