# OpenReview forum: "Dissecting Recall of Factual Associations in Auto-Regressive Language Models"
_EMNLP/2023/Conference — EMNLP 2023 Main_

### Official Review · Reviewer_cRFb · 2023-08-04

**Soundness:** 4

**Excitement:**

4: Strong: This paper deepens the understanding of some phenomenon or lowers the barriers to an existing research direction.

**Paper Topic And Main Contributions:**

The paper studies how transformer-based language models retrieve information to predict for subject-relation queries. The authors designed methods to study the effects on prediction by blocking updates in certain components (e.g., attention edges, MLP layers), and present some interesting and useful results (e.g., information flows through certain critical points to prediction, attribute information was gradually built up in the representation at the last subject position through layers).

**Reasons To Accept:**

Overall the paper is well written and easy to understand. It presents interesting and useful findings for better understanding how a transformer-based LM predicts for subject-relation query. For example, they found that early layers are important in propagating information for predicting the attribute. Another example is the attribute information is aggregated gradually through layers. These findings could not only help understand the models but also inspire new methods that better aggregate information for subject-relation queries.  The experiments are comprehensive and provide answers for questions at different levels.




**Reasons To Reject:**

The study is only on subject-relation query. It might not be easy to apply the proposed methods to understand the information flow in other research tasks, where important words in query are not known beforehand (just as the subject and relation word in the subject-relation query task), such as reading comprehension.

**Reproducibility:**

4: Could mostly reproduce the results, but there may be some variation because of sample variance or minor variations in their interpretation of the protocol or method.

**Reviewer Confidence:**

3: Pretty sure, but there's a chance I missed something. Although I have a good feel for this area in general, I did not carefully check the paper's details, e.g., the math, experimental design, or novelty.

---

> ### Author Rebuttal · Authors · 2023-08-28
>
> We thank the reviewer for carefully reading our work and for the positive review!
>
> Regarding our focus on subject-relation queries (**"The study is only on subject-relation query"**) – this is an important point that we address below, thanks for raising this.
>
> First, it is important to emphasize that the goal of our work is to obtain a deep and mechanistic understanding of the recall process of facts in LMs. We believe that, **given the high complexity of modern LMs, the path towards this goal must start from basic and simple settings**. Notably, many insightful recent works have been following this intuition as well, and analyzed tasks of equal complexity (see example references [1], [2], [3], [4] below).
>
> Nonetheless, we agree that moving to more complex tasks is very interesting and important. However, such complex inputs are likely to involve *additional* mechanisms, such as multi-hop reasoning, extraction of intermediate subject-object-relation tokens, and coreference resolution. Investigating each of these internal mechanisms can be done in isolation and then together. Therefore, we believe that **our work sets the ground for such future analyses**, since it reveals the basic mechanisms of knowledge recall (that are expected to occur in more complex inputs) and contributes practical methods that can be applied as-is or easily adjusted for such inputs. For instance, knocking out multiple positions at once could be used to study the hierarchy through which LMs process predictions, and the attribute-rate approximation can be applied to any hidden representation during inference.
>
> &emsp;
>
> ___
>
>
> [1] Interpretability in the Wild: a Circuit for Indirect Object Identification in GPT-2 Small. Wang et al., ICLR, 2023.
>
> [2] Locating and Editing Factual Associations in GPT. Meng et al., NeurIPS 2022.
>
> [3] In-context Learning and Induction Heads. Olsson et al., 2022.
>
> [4] Understanding Arithmetic Reasoning in Language Models using Causal Mediation Analysis. Stolfo et al., 2023.

---

### Official Review · Reviewer_6DKT · 2023-08-04

**Soundness:** 4

**Excitement:**

4: Strong: This paper deepens the understanding of some phenomenon or lowers the barriers to an existing research direction.

**Paper Topic And Main Contributions:**

The paper explores the how factual information is retrieved from transformer-based LMs. The authors perform a number of experiments to reveal some unexpected results regarding the location of factual associations. The work suggests further research directions in the form of knowledge localization and model editing.

**Reasons To Accept:**

With the rise of the desire for explainability of LMs works which identify (or go some way towards identifying) important locations are of great potential value.

The paper uses figures to advantage, illustrating complex concepts and results.


**Reasons To Reject:**

This is a fairly technical paper, presumably a stepping stone on an incremental path.

**Reproducibility:**

2: Would be hard pressed to reproduce the results. The contribution depends on data that are simply not available outside the author's institution or consortium; not enough details are provided.

**Reviewer Confidence:**

1: Not my area, or paper was hard for me to understand. My evaluation is just an educated guess.

---

> ### Author Rebuttal · Authors · 2023-08-28
>
> We thank the reviewer for putting an effort into reviewing our work, despite that it is not their area of expertise, and for appreciating the potential value in works of this kind.
>
> We address the reviewer’s point that **"This is a fairly technical paper, presumably a stepping stone on an incremental path"**:
>
> We share the reviewer’s view that understanding how Transformers (and neural networks in general) work will not happen overnight. Nonetheless, as research on interpretability is growing fast, it provides fundamental insights that allow developing various methods to increase transparency and control over LMs, and set the groundwork for devising better architectures.
>
> We believe that our work is indeed a stepping stone towards this larger goal, since it analyzes one of the most fundamental questions about LMs – how they represent and recall knowledge from their parameters. Answering this question could have far-reaching consequences on many central problems with LMs today, such as hallucinations, memorization, and biases. Our work makes substantial contributions in this context; it reveals the basic stages of knowledge recall and identifies on a more fine-grained level where factual associations are stored and they are extracted during inference, using (novel) attention knockout and vocabulary projection techniques.

---

### Official Review · Reviewer_Bdt7 · 2023-08-05

**Typos Grammar Style And Presentation Improvements:** L327 Attributes Rate --> Attribute Rate
**Soundness:** 4

**Excitement:**

3: Ambivalent: It has merits (e.g., it reports state-of-the-art results, the idea is nice), but there are key weaknesses (e.g., it describes incremental work), and it can significantly benefit from another round of revision. However, I won't object to accepting it if my co-reviewers champion it.

**Paper Topic And Main Contributions:**

This paper proposes to use subject-relation queries to investigate how Transformer-based auto-regressive language models (GPT-2 and GPT-J) aggregate information about the subject and relation to predict the correct attribute.

The attention "knockout" method is applied to identify critical information flow points in factual predictions. The authors argue that critical information from the subject positions moves directly to the last position at middle-upper layers.

"Attributes Rate" is used to investigate the position of subject representation. The authors find that the model constructs attribute-rich subject representations at the last subject-position. Attribute rates of token embeddings (from the embedding layer?) are also studied, and it reveals that while static subject-token embeddings encode some factual associations, other model components are needed for extraction of subject-related attributes. It is also found that canceling the early MLP sublayers has a destructive effect on the subject representation’s attributes rate, while canceling early MHSA sublayers does not have such a strong effect.

"Extraction Rate" is used to investigate how and where attributes are extracted. The authors suggest that both the MHSA and MLP implement attribute extraction, but MHSA is the prominent mechanism for factual queries. It is also indicated that subject enrichment, through which the model constructs a representation at the last subject-position that encodes many subject-related attributes, has a big impact. Overall, factual associations are encoded in the MHSA parameters, acting as “knowledge hubs”.

**Reasons To Accept:**

1. The study of internal representations of Transformer-based models is still an evolving domain, which needs more contributions. Particularly, little is known about how factual predictions are built.
2. This paper proposes an automatic approximation of the subject-attribute relatedness, namely the attribute rate. It is a reasonable and novel idea.
3. The experiments are well designed and the results are well presented. There are some interesting findings.

**Reasons To Reject:**

1. Although we should start with simple experiments in order to study a complex system, it is often unclear whether conclusions drawn from simple experiments can extend to general, complicated cases. It is difficult to tell whether other NLP tasks follow the same pattern.
The authors suggest that "these findings open new research directions for knowledge localization and model editing", but it is unclear whether the localization depends on training tasks, or model size, among other things.

**Reproducibility:**

3: Could reproduce the results with some difficulty. The settings of parameters are underspecified or subjectively determined; the training/evaluation data are not widely available.

**Reviewer Confidence:**

3: Pretty sure, but there's a chance I missed something. Although I have a good feel for this area in general, I did not carefully check the paper's details, e.g., the math, experimental design, or novelty.

---

> ### Author Rebuttal · Authors · 2023-08-28
>
> We thank the reviewer for the thorough review, and are encouraged that they find our findings interesting, our experiments to be well designed, and the results to be well presented!
>
> We address the points raised in the "Reasons to Reject" section:
>
>
> >**"... it is often unclear whether conclusions drawn from simple experiments can extend to general, complicated cases. It is difficult to tell whether other NLP tasks follow the same pattern"**
>
> First, it is important to emphasize that the goal of our work is to obtain a deep and mechanistic understanding of the recall process of facts in LMs. We believe that, **given the high complexity of modern LMs, the path towards this goal must start from basic and simple settings**. Notably, many insightful recent works have been following this intuition as well, and analyzed tasks of equal complexity (see example references [1], [2], [3], [4] below).
>
> Nonetheless, we agree that moving to more complex tasks is very interesting and important. However, such complex inputs are likely to involve *additional* mechanisms, such as multi-hop reasoning, extraction of intermediate subject-object-relation tokens, and coreference resolution. Investigating each of these internal mechanisms can be done in isolation and then together. Therefore, we believe that **our work sets the ground for such future analyses**, since it reveals the basic mechanisms of knowledge recall (that are expected to occur in more complex inputs) and contributes practical methods that can be applied as-is or easily adjusted for such inputs. For instance, knocking out multiple positions at once could be used to study the hierarchy through which LMs process predictions, and the attribute-rate approximation can be applied to any hidden representation during inference.
>
> &ensp;
>
> >**"... it is unclear whether the localization depends on training tasks, or model size, among other things"**
>
> Thank you for this comment. While we agree that it is not obvious that our findings will generalize across all these factors, we believe that our choice of models and the correspondence of our findings with those of other works that analyzed other models, provide a good basis to expect that our analysis, tools, and observations are general and robust. We explain this in detail next.
>
> First, the models we analyzed have **the same basic architecture that underlies most LMs today, yet, they are different in many factors including size, layout (depth vs. width), transformer block configuration (sequential vs. parallel MLP layers), and training data.**  Therefore, these models represent a large family of modern auto-regressive LMs. Specifically, LLaMA, Alpaca, Vicuna, GPT-NeoX, Pythia, OPT and others, all rely on the same variations analyzed in our work. The fact that we observed similar trends in the models we analyzed is not trivial and makes a good basis to believe that these results will generalize to other auto-regressive decoder-only LMs. Of course, there might be some variations in the specific range of layers, but this is not the purpose of this analysis; rather it is the stages of the input processing and the role of the different modules (MHSA/MLP) in that process.
>
> In addition, our analysis methods leverage techniques, such as projection to the vocabulary and interventions, that were successfully applied to a wide range of models, including models such as BERT and T5 [5][6][7] that were trained with different objectives of masked language modeling and contrastive learning for dense passage retrieval. Therefore, **one could build up on our work and easily apply our methods to study factual recall in other models**. We agree that this is a very interesting question to explore in future work.
>
> Last, many knowledge localization and editing works have used the same specific models we analyzed to develop editing methods, such as ROME [2], MEMIT [8], and MEND [9]. Hence, our findings provide further insights to understand their success and failure cases, potentially leading to better methods.
>
> &ensp;
> ___
>
> [1] Interpretability in the Wild: a Circuit for Indirect Object Identification in GPT-2 Small. Wang et al., ICLR, 2023.
>
> [2] Locating and Editing Factual Associations in GPT. Meng et al., NeurIPS 2022.
>
> [3] In-context Learning and Induction Heads. Olsson et al., 2022.
>
> [4] Understanding Arithmetic Reasoning in Language Models using Causal Mediation Analysis. Stolfo et al., 2023.
>
> [5] Understanding Transformer Memorization Recall Through Idioms. Haviv et al., EACL, 2023.
>
> [6] What Are You Token About? Dense Retrieval as Distributions Over the Vocabulary. Ram et al., ACL, 2023.
>
> [7] Neural Knowledge Bank for Pretrained Transformers. Dai at al., 2022.
>
> [8] Mass Editing Memory in a Transformer. Meng et al., ICLR, 2023.
>
> [9] Fast Model Editing at Scale. Mitchell et al., ICLR, 2022.

---

### Meta-Review · Area_Chair_hsmQ · 2023-09-17

**Recommendation:** 5

**Metareview:**

The paper provides a well-presented study on how factual associations flow through the internal structure of a transformer. It addresses an important topic and includes relevant findings. It is a targeted study with a (narrow) focus on subject-query, that can be seen as a nice first step into diving further into this topic.

---

### Decision · Program_Chairs · 2023-10-07

**Decision:**

Accept-Main

**Comment:**

The paper provides a well-presented study on how factual associations flow through the internal structure of a transformer. It addresses an important topic and includes relevant findings. It is a targeted study with a (narrow) focus on subject-query, that can be seen as a nice first step into diving further into this topic.